# Evaluation of Cytotoxic and Mutagenic Effects of the Synthetic Cathinones Mexedrone, α-PVP and α-PHP

**DOI:** 10.3390/ijms22126320

**Published:** 2021-06-12

**Authors:** Monia Lenzi, Veronica Cocchi, Sofia Gasperini, Raffaella Arfè, Matteo Marti, Patrizia Hrelia

**Affiliations:** 1Department of Pharmacy and Biotechnology, Alma Mater Studiorum University of Bologna, 40126 Bologna, Italy; m.lenzi@unibo.it (M.L.); veronica.cocchi4@unibo.it (V.C.); sofia.gasperini4@unibo.it (S.G.); patrizia.hrelia@unibo.it (P.H.); 2Section of Legal Medicine and LTTA Centre, Department of Translational Medicine, University of Ferrara, 44121 Ferrara, Italy; raffaella.arfe@unife.it; 3Collaborative Center for the Italian National Early Warning System, Department of Anti-Drug Policies, Presidency of the Council of Ministers, 44121 Ferrara, Italy

**Keywords:** synthetic cathinones, mexedrone, α-PVP, α-PHP, mutagenicity, S9 mix, ROS, flow cytometry, novel psychoactive substances

## Abstract

Mexedrone, α-PVP and α-PHP are synthetic cathinones. They can be considered amphetamine-like substances with a stimulating effect. Actually, studies showing their impact on DNA are totally absent. Therefore, in order to fill this gap, aim of the present work was to evaluate their mutagenicity on TK6 cells. On the basis of cytotoxicity and cytostasis results, we selected the concentrations (35–100 µM) to be used in the further analysis. We used the micronucleus (MN) as indicator of genetic damage and analyzed the MNi frequency fold increase by flow cytometry. Mexedrone demonstrated its mutagenic potential contrary to the other two compounds; we then proceeded by repeating the analyzes in the presence of extrinsic metabolic activation in order to check if it was possible to totally exclude the mutagenic capacity for α-PVP and α-PHP. The results demonstrated instead the mutagenicity of their metabolites. We then evaluated reactive oxygen species (ROS) induction as a possible mechanism at the basis of the highlighted effects but the results did not show a statistically significant increase in ROS levels for any of the tested substances. Anyway, our outcomes emphasize the importance of mutagenicity evaluation for a complete assessment of the risk associated with synthetic cathinones exposure.

## 1. Introduction

The number of novel psychoactive substances (NPS) newly introduced to the European market over the last years has risen sharply. The dramatic increase of NPS on the illicit drug market, with the aim of bypassing the controlled substances legislation, is a public health and regulatory challenge of worldwide concern. NPS are often almost automatically associated with phenomena such as addiction, intoxication and overdose, that can lead to hospitalisation or even death.

To date, most of the available toxicological information essentially derive from acute or short-term observations, while the long-term effects seem to be nearly unknown.

To fill this gap, for a couple of years we have been carrying out several studies about numerous NPS belonging to different classes. The aim was, in particular, to investigate their possible mutagenicity, whose key role in the potential long-term toxicity is well known.

We therefore published a first study in 2020 in which it was possible to demonstrate for the first time the mutagenic capacity, in terms of increasing the frequency of MNi, of four synthetic cannabinoids, STS-135, APINAC, JWH-018-Cl and BB-22, and confirm what was reported by Koller et al. for JWH-018 and 5F-AKB-48 both in terms of cytotoxicity and cystostasis and in terms of MNi frequency [1,2,3]. 

This important result further supported the need to implement the available data and to analyze other classes of NPS. Indeed, this could help to understand if a class toxicity characteristic for synthetic cannabinoids could be defined and therefore if it could be connectable to a particular chemical structure. For this reason, we decided to address the evaluation of the mutagenic potential of psychedelic molecules, in particular focusing on four psychoactive phenethylamines, 2C-H, 2C-I, 2C-B e 25B-NBOMe [4]. Additionally, in this case, it was possible to demonstrate how all substances proved to be mutagen and to identify a possible involvement of oxidative stress as a mechanism underlying the genetic damage highlighted. These results suggested us the next natural step, that was to continue the studies by investigating the cytotoxic and mutagenic potential and the ability to induce oxidative stress of synthetic cathinones, molecules that mainly dominated seizures in Europe in 2020 [5]. A small but increasing number of synthetic cathinones production sites has been discovered [5]. Synthetic cathinones are chemical analogues of a natural stimulant (cathinone) found in the leaves of a psychoactive plant [6]. Synthetic analogues of cathinone, also called β-keto amphetamines, are chemical structures that can be easily modified leading to a wide range of psychostimulant compounds. Even though the law has evolved to schedule NPS and limit their spread, the overall availability of new synthetic cathinones has not decreased. Among these, mexedrone (3-methoxy-2-(methylamino)-1-(4-methylphenyl)propan-1-one), α-PVP (alpha-pyrrolidinopentiophenone) and α-PHP (alpha-pyrrolidinohexanophenone), analogously to other cathinones, have been associated with fatal and non-fatal intoxications emphasizing the importance of knowing of their pharmaco-toxicological profile [7,8,9,10,11,12,13,14,15]. α-PVP, and its less common analogue α-PHP, are pyrovalerone type designer drugs. Pyrovalerone derivatives (α-pyrrolidinophenones) are a synthetic cathinones subgroup chemically characterized by a pyrrolidine ring and a long side chain linked to the α-carbon [10]. α-PVP was first detected in the EU in 2011, while the first report on α-PHP identification in Japan was published in 2014 [7,16]. Mexedrone was identified by Qian and collaborators in 2016 and is a lesser-known alpha-methoxy derivative of the popular synthetic cathinone mephedrone (4-methyl-N-methyl cathinone), a drug linked to neurotoxicity [17,18,19]. The abuse of these substances is related to their psychostimulant effects that are similar to those induced by cocaine, amphetamine and MDMA [6]. Mexedrone, α-PVP and α-PHP have been frequently linked to polydrug use cases and, as with other synthetic cathinones, induce effects such as increased energy, reduced appetite, euphoria, sociability, and intensified sensory experiences [10,11]. In addition to this typical stimulant profile, different adverse effects such as tachycardia, restlessness, agitation, bruxism, seizures, hallucinations, and psychosis should be considered [8,9,11,20,21,22,23]. At the molecular level, both effects and adverse effects of these novel recreational drugs can be related to their interaction with dopamine (DAT), norepinephrine (NET), and serotonin (SERT) transporters [24,25,26,27]. Particularly, pyrovalerone derivatives pharmacological activity has been recently studied in vitro using human embryonic kidney 293 cells. α-PVP (DAT IC_50_ = 0.04 µM; NET IC_50_ = 0.02 µM; SERT IC_50_ > 10 µM) and α-PHP (DAT IC_50_ = 0.02 µM; NET IC_50_ = 0.04 µM; SERT IC_50_ > 10 µM) selectively and very potently inhibit the dopamine and norepinephrine reuptake with negligible affinity for the SERT and do not enhance neurotransmitters release, as opposed to other synthetic cathinones [24,25]. Differently, a recent in vitro study in rats’ brain synaptosomes has shown that mexedrone (DAT IC_50_~6.8 µM; NET IC_50_~8.8 µM; SERT IC_50_~5.2 µM) weakly inhibit monoamines uptake and exert a weak releasing activity on SERT (EC_50_ = 2.5 µM) [28]. Among all available studies, the bibliographic searches conducted on the main databases did not report any genotoxicological investigation concerning mexedrone, α-PVP and α- PHP and only one regarding 3-MMC and 4-MEC is present [29].

For these reasons, we evaluated in vitro, on human lymphoblastoid TK6 cells, the mutagenic potential of these three synthetic cathinones, in particular in terms of MNi frequency fold increase, following the OECD guideline n° 487 [30] and using an automated version by flow cytometry (FCM) recently developed [31].

## 2. Results

In the preliminary phase of the research, we determined the concentrations to be used in the subsequent experiments aimed at evaluating the potential mutagenicity of synthetic chatinones in study. First, we assessed the mexedrone, α-PVP and α-PHP induced cytotoxicity after 26 h treatment by measuring the percentage of live cells at the different concentrations tested (25, 35, 50, 75, 100 µM). This value was normalized on the one obtained in the concurrent negative control cultures in order to check that the cellular viability complied with the OECD threshold (equal to 55 ± 5%) [30].

In Figure 1 it can be seen how the viability is abundantly above the OECD threshold (represented by the red line) for all substances (Figure 1A,B,C).

In addition, to make the mutagenicity test reliable, it is necessary to check the cellular proliferation in order to verify that a sufficient number of cells has undergone mitosis and so that they are able to transmitt the genetic damage suffered to the daughter cells. For this purpose, the OECD recommends the measurement of the Relative Population Doubling (RPD) to estimate the cytostasis and, analogously to the cytotoxicity, establishes a threshold at most equal to 55 ± 5% [30]. Similarly, to what has been reported for cytotoxicity, all three substances under study, at all the concentrations tested, showed an RPD well above the established threshold (Table 1).

The research continued by considering another alternative cell death mechanism, i.e., apoptosis, since the assessment of other cytotoxicity markers could provide useful additional information (e.g., cell integrity, apoptosis, necrosis, etc.).

The evaluation of the apoptotic process was carried out according to the Guava Nexin assay. For mexedrone, α-PVP and α-PHP the induction of apoptosis never reached a doubling if compared to the concurrent negative control at all the selected concentrations (Figure 2A,B,C).

Overall, the obtained results allowed selecting the concentrations to be used in the mutagenic analysis. In particular, 75 and 100 µM were tested for all synthetic cathinones. The cultures treated in this way, were compared with negative and positive (Mytomicin (MMC) or Vinblastine (VINB)) treated cultures.

In particular, mexedrone showed a significant increase of the MNi frequency at both the concentrations tested (75 and 100 µM) (Figure 3); while for α-PVP (Figure 4) and α-PHP (Figure 5) no statistically significant increase was observed.

Subsequently, given the absence of mutagenicity of α-PVP and α-PHP, we analyzed the possible mutagenic activity of the metabolites produced by all three synthetic cathinones in study.

In particular, the cultures were treated with 75 and 100 µM concentrations and then were compared to negative and positive (cyclophosphamide (CP)) controls. Mexedreone showed a statistically significant increase at both concentrations tested similarly to the results obtained in the absence of S9 mix (Figure 6). α-PVP showed a significant increase in MNi frequency at both concentrations tested (Figure 7), while for α-PHP a statistically significant increase was observed only at the highest concentration tested (Figure 8).

Lastly, in order to identify a possible mechanism of action at the basis of the mutagenic activity, TK6 cells were treated with the synthetic cathinones for 1, 6 and 12 h and then the possible ROS induction was measured.

The highest concentration tested was selected for each substance: 100 µM for all cathinones. The cultures treated in this way were compared to negative and positive (H_2_O_2_) controls. The results did not show a statistically significant increase of the mean fluorescence intensity if compared to the concurrent negative control for all the cathinones in study (Figure 9).

## 3. Discussion

To date, studies involving mutagenicity and the potential long-term effects of mexedrone, α-PVP and α-PHP are totally absent, so we proceeded to their cytotoxic and mutagenic evaluation. This research was entirely conducted in vitro on TK6 cells, selected on the basis of their human and non-tumoral origin, among cell lines recognized by OECD as validated assay systems for mutagenic evaluation [30]. The tests were all carried out by FCM which, in our two previous studies, has already proved to be a very effective platform to screen multiple molecules at the same time, as it allows an objective multi-parameter measurement supported by a high number of events analyzed in a very short time [1,4]. In particular, for the evaluation of mutagenic potential an innovative protocol was also used, which has been recently published and developed in our laboratory for the execution in FCM of the “*In vitro mammalian cell micronucleus test*”, corresponding to OECD guideline no. 487 [30]. Indeed, although MN has long been recognized as a valid biomarker of chromosomal damage and genomic instability, the classical analysis method in optical microscopy shows some inevitable criticalities, including long sample preparation and analysis times and the subjectivity of interpretation by the operator.

In order to proceed with the evaluation of mutagenicity, it is necessary that the treated population has at least 40% of live and proliferating cells if compared to the control cultures. Therefore, in the first phase of the research, aliquots of TK6 cells were treated with the three synthetic cathinones under study in different concentrations for 26 h, corresponding to the time required for TK6 cells to perform two replication cycles. At the end of the treatment, cell viability and proliferation were evaluated. The Guava ViaCount assay showed that mexedrone, α-PVP and α-PHP do not induce any cytotoxic or cytostatic effects up to the highest concentration tested (100 μM).

At the same time, apoptosis was also evaluated, as a cell death mechanism alternative to necrosis and particularly important for the purpose of mutagenicity. Indeed, the population of cells that has suffered irreparable genetic damage could be selectively eliminated by means of apoptosis. The results obtained with the Guava Nexin assay allowed highlighting a similar behaviour for all the three molecules analyzed. In particular, at all the concentrations tested, mexedrone, α-PVP, and α-PHP did not increase the percentage of apoptotic cells at all. This has important repercussions in terms of mutagenicity because it highlights the cell’s inability to counteract, through this mechanism of programmed death, the transmission of an aberrant genome to daughter cells.

Overall, the results obtained in these assays permitted to define the concentrations to be tested in the subsequent mutagenicity studies. Then, the automated flow cytometric protocol allowed demonstrating the ability of mexedrone to increase the frequency of MNi in a statistically significant manner and therefore to prove itself to be mutagenic, in particular in terms of induction of chromosomal aberrations. In contrast, for α-PVP, and α-PHP no increase in the frequency of MNi was observed. The behaviour exhibited by mexedrone agrees with what Al-Serori et al. demonstrated for 3-MMC and 4-MEC [29] and, when compared to α-PVP and α-PHP, underlines how, not only among NPS belonging to different classes but also within the same class, molecules that share a strong structural analogy may have very different toxicological spectra.

However, the deficiency of mutagenic capacity of α-PVP and α-PHP could only be apparent and due to the poor metabolic capacity of the cell line employed in this study. Indeed, as previously pointed out, TK6 cells are recognized by OECD as the validated assay system for in vitro mutagenesis evaluation, but it has been shown that they have negligible expression of the main cytochrome CYP-450 isoforms, which are responsible for the metabolic transformation of chemical substances [32].

To overcome this lack, the guidelines suggest the use of an extrinsic metabolic activation system. Therefore, the study continued in this direction i.e., also investigating the possible mutagenic activity of the metabolites produced by mexedrone, α-PVP and α-PHP. For this purpose, S9 mix has been added to the cultures, then treated with the synthetic cathinones for a short treatment time (3 h) followed by a recovery period in fresh medium of 23 h, as recommended by OECD 487 [30]. The results, albeit preliminary, showed a statistically significant increase in MNi frequency for all synthetic cathinones, allowing us in particular to hypothesize a possible mutagenic activity of the metabolites produced by α-PVP and α-PHP. This result confirms the well-known role of metabolism in transforming the so-called pre-mutagen into final mutagen, generating metabolites with chemical and structural characteristics such as to be able to interact with DNA, form a stable bond and consequent damage which, if not properly repaired, it will be fixed and will result in an irreversible mutation. The fact that α-PVP and α-PHP produce metabolites endowed with mutagenic capacity allows ascribing also to these two synthetic cathinones the ability to induce chromosomal aberrations, an aspect that further aggravates their toxicity profile, particularly in the long term, as it is well known the close link between mutations and chronic-/neuro-degenerative pathologies.

Although knowing the pharmacokinetic profile of these synthetic cathinones could be fundamental to predict and better evaluate their potential toxic properties, very little is known about it. Recent in vitro studies have reported the identification of 6 phase I metabolites of α-PVP and showed PVP lactam and ß-hydroxy PVP as major metabolites [33,34,35]. It has also been reported that 19 phase I metabolites and 9 glucuronide conjugates of α-PHP were identified in human urine of a drug abuser. Among these, β-keto-reduced alcoholic dihydro metabolite and glucuronidated ß-keto-reduced alcohols were the most represented and a recent study has identified α-PHP dihydroxy-pyrrolidinyl as additional major metabolite [36,37,38]. Differently, although its synthetic pathways have been recently investigated, mexedrone pharmacokinetic profile is still unclear [28].

In addition to identification, it would be interesting also to isolate the metabolites produced in vitro and test them individually in the same way employed in the present work, in order to identify which and/or how many of these metabolites are responsible for the mutagenic effect. However, the really fundamental result, already obtained from our preliminary studies, is the demonstrated mutagenic capacity of all three synthetic cathinones under study, regardless of its attribution to the parental compound or one of its metabolites.

A due consideration must be made also on oxidative stress induction. Our study ended by analysing the possible formation of ROS after 1, 6, 12 h treatment with the three synthetic cathinones under study, in order to investigate one possible mechanism at the basis of the mutagenic activity demonstrated. It is indeed well known how ROS, such as ^1^O_2_, O_2_∙−, H_2_O_2_, ∙OH, are involved in genetic damage. Our results revealed no statistically significant increase in ROS levels at any of the treatment times and for all the synthetic cathinones analyzed, suggesting, in particular, the involvement of other mechanisms underlying the proven damage to the genetic material. It is important to point out that the results we obtained for α-PVP differ from what is reported in the literature. A possible explanation lies in the fact that Zhou et al. showed a statistically significant increase in mitochondrial superoxide production by treating a different cell line (C2C12 myoblasts) for a definitely longer time (24 h) and with concentrations from 10 to 20 times higher than ours [39]. Analogously, the ability to induce oxidative stress demonstrated, mainly on neuronal cells, also by other synthetic cathinones, could be similarly justified by the high doses tested and the long treatment times [40,41].

An interesting aspect to think about could be the relevance in vivo of our in vitro results. Regarding in particular the proved mutagenicity of the molecules under study, it is important to underline that our findings agree with the mutagenicity demonstrated for mephedrone [42] and for khat extract [43] in rats and for cathinone in mice [44,45]. Moreover, it could be asked whether the concentrations we tested are reachable in vivo and if they are comparable to the level of synthetic cathinones detectable in human users. However, it is recognised that it is not possible to define a No-Observed-Adverse-Effect Level (NOAEL) for genotoxic substances and zero risk corresponds only to the zero dose, therefore, any dose is potentially toxic. increasing the dose and the exposure, the likelihood that damage will occur increases [46].

The research carried out in the present work does not provide all the answers but adds another small piece to the puzzle of the toxicological knowledge of synthetic cathinones and rather stresses the need to continue the studies, testing both the parental compounds and the corresponding metabolites.

## 4. Materials and Methods

### 4.1. Reagents

Ethylenediaminetetraacetic acid (EDTA), fetal bovine serum (FBS), L-glutamine (L-Gln), Mitomycin C (MMC), Nonidet, penicillin-streptomycin solution (PS), phosphate buffered saline (PBS), cyclophosphamide, potassium chloride, potassium dihydrogen phosphate, Roswell Park Memorial Institute (RPMI) 1640 medium, water bpc grade, ethanol, sodium chloride, sodium hydrogen phosphate, vinblastine (VINB), 2′-7′-dichlorodihydrofluorescin diacetate (DCFH-DA) (all purchased from Merck, Darmstadt, Germany), Guava Nexin Reagent, Guava ViaCount Reagent (all purchased from Luminex Corporation, Austin, TX, USA), RNase A, SYTOX Green, 7-aminoactinomycin D (7-AAD) (purchased from Thermo Fisher Scientific, Waltham, MA, USA), Mutazyme 10% S9 Mix (purchased from Moltox, NC, United States).

### 4.2. Synthetic Cathinones

The synthetic cathinones mexedrone, α-PVP, α-PHP were purchased from LGC Standards (LGC Standards S.r.L., Sesto San Giovanni, Milan, Italy) and www.chemicalservices.net (accessed on 12 June 2021).

All synthetic cathinones were dissolved in absolute ethanol up to 10 mM stock solution and stored at −20 °C. Absolute ethanol concentration was always in the range 0.25–1.0% in all experimental conditions, to avoid potential solvent toxicity.

### 4.3. Cell Culture and Treatments

Human TK6 lymphoblast cells were purchased by Merck (Darmstadt, Germany) and were grown at 37 °C and 5% CO_2_ in RPMI-1640 supplemented with 10% FBS, 1% L-GLU and 1% PS. To maintain exponential growth and considering that the time required to complete the cell cycle is 13–14 h, the cultures were divided every three days in fresh medium and the cell density did not exceed the critical value of 9 × 10^5^ cells/mL.

In all experiments, aliquots of 2.5 × 10^5^ of TK6 cells in 1 mL were seeded and treated with increasing concentrations of synthetic cathinones from 0 to 100 µM for 26 h without S9 Mix or for 3 h followed by 23 h recovery period with or without S9 Mix.

Cytotoxicity, cytostasis and apoptosis were measured after 26 h of treatment while mutagenicity at both treatment times.

Finally, the analysis of ROS was performed after 1, 6 and 12 h of treatment.

### 4.4. Flow Cytometry

All FCM analysis reported below were performed using a Guava easyCyte 5HT flow cytometer equipped with a class IIIb laser operating at 488 nm (Luminex Corporation, Austin, TX, USA).

### 4.5. Cytotoxicity Analysis

Cytotoxicity assay was performed as previously described by Lenzi et al. [1,2] and Cocchi et al. [3]. Briefly, at the end of the treatment time, 1.5 × 10^5^ cells were stained with 180 µL of Guava ViaCount Reagent and incubated for 5 min at room temperature (RT) The reagent allows the discrimination of viable from dead cells based on their different permeability to DNA-binding dye. A total of 1000 events were acquired and analyzed by Guava ViaCount software.

The results obtained from the samples treated with different concentrations of synthetic cathinones were normalized on those obtained from the untreated control cultures, accounted equal to 100%. These results were used to check that the cellular viability percentage respected the OECD threshold (55 ± 5%) for each synthetic cathinones treatment [30].

### 4.6. Cytostasis Analysis

In parallel, always using the Guava ViaCount Reagent, the number of cells seeded at time 0 and the one measured at the end of the treatment time were used to check the correct replication and to calculate the Population Doublings (PD) values. Subsequently, the PD obtained in the control cultures was compared to that measured in the treated cultures obtaining the Relative Population Doublings (RPD). PD and RPD were calculated with the following formulas:(1)$$PD=\log\left(\frac{post-initial\;cell\;number}{initial\;cell\;number}\right)÷\log2$$
(2)$$RPD=\frac{PD\;in\;treated\;cultures}{PD\;in\;control\;cutures}\;x\;100$$

Similarly, to cytotoxicity, the cytostasis was checked, in order to verify that cell proliferation respected the threshold established by OECD guideline (55 ± 5%) [30].

### 4.7. Apoptosis Analysis

The percentage of apoptotic cells was evaluated by the Guava Nexin Assay according to the procedure by Lenzi et al. [1,31] and Cocchi et al. [4].

Briefly, at the end of the treatment time the percentage of apoptotic cells was assessed using the Guava Nexin Reagent. The reagent contains the 7-aminoactinomycin D (7- AAD), a cells non-permeant dye that allows the discrimination between live and dead cells, and the Annexin-V-PE that allows the identification of apoptotic cells by binding to the phosphatidylserine on the cell surface.

100 μL of reagent was added to 100 μL of cell suspension (~1 × 10^5^ cells) and incubated for 20 min at RT. Then, a total of 2000 events were acquired and analyzed by Guava Nexin software.

The apoptotic cell percentages recorded in the cultures treated with synthetic cathinones different concentrations were normalized on those recorded in the untreated control cultures, accounted equal to 1 and expressed as apoptotic fold increase. These results were used to check that the apoptosis induction for each synthetic cathinone was similar or corresponding at most to a doubling of that recorded in the untreated control cultures [1,4,31].

### 4.8. Genotoxicity Analysis

The analysis of the MNi frequency was performed using an automated protocol published by Lenzi et al. [31].

Briefly, at the end of the treatment time 7 × 10^5^ cells were collected, incubated with 7-AAD at RT for 5 min, lysed and stained with SYTOX Green. A total of 10,000 nuclei, derived from viable and proliferating cells, and the corresponding number of MNi were acquired and analyzed by Guava Incyte software. The discrimination between nuclei and MNi occurs on the basis of the different size and intensity of green fluorescence.

The MNi frequency (number of MNi/10,000 nuclei) recorded in treated cultures were normalized on those recorded in the concurrent negative control cultures, accounted equal to 1 and expressed as MNi frequency fold increase.

The exogenous metabolic activation system employed is a co-factor-supplemented post-mitochondrial fraction (S9 mix) prepared from the livers of rats treated with enzyme-inducing agent Aroclor 1254. The final concentration of S9 in the culture medium was 0.75%.

We used the clastogen MMC and the aneuploidogen VINB as positive controls in absence of S9 mix, while CP, a clastogen agent requiring metabolic activation, in presence of S9 mix [30].

### 4.9. ROS Analysis

The analysis of ROS levels was performed using the dichlorofluorescin assay as previously described by Cocchi et al. [4].

Briefly, at the end of the treatment time 2 × 10^5^ cells were stained at 37 °C in the dark for 20 min with 2′,7′-dichlorodihydrofluorescin diacetate (DCFH2-DA). A total of 5000 events, derived from viable cells, were acquired and analyzed by Guava Incyte software.

The fluorescence intensity of 2′,7′-dichlorofluorescin (DCF) (which is formed in cells in presence of ROS) recorded in treated cultures were normalized on those recorded in the untreated control cultures, accounted equal to 1 and expressed as ROS fold increase.

We used H_2_O_2_ as positive control.

### 4.10. Statistical Analysis

All results were expressed as mean ± SEM of at least five independent experiments. All the statistical analyzes were performed using the Analysis of Variance for paired data (repeated ANOVA), followed by Dunnett or Bonferroni as post-tests (Prism Software 4).

## Figures and Tables

**Figure 1 ijms-22-06320-f001:**
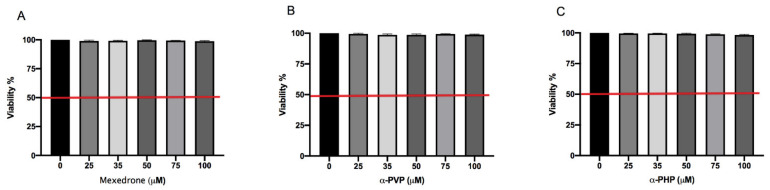
Cell viability on TK6 cells after 26 h treatment with mexedrone (**A**), α-PVP (**B**), α-PHP (**C**) at the indicated concentrations compared to the negative control [0 μM]. Each bar represents the mean ± SEM of five independent experiments. Data were analyzed by ANOVA Repeated followed by Dunnet post-test.

**Figure 2 ijms-22-06320-f002:**
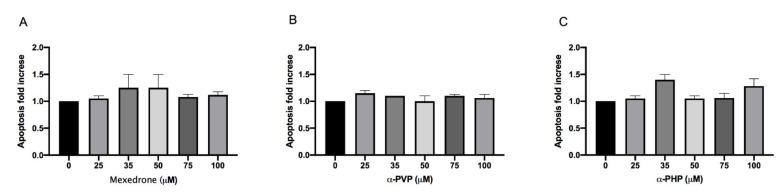
Apoptosis fold increase on TK6 cells after 26 h treatment with mexedrone (**A**), α-PVP (**B**) and α-PHP (**C**) at the indicated concentrations compared to the negative control [0 μM]. Each bar represents the mean ± SEM of five independent experiments. Data were analyzed using repeated ANOVA followed by Dunnet post-tests.

**Figure 3 ijms-22-06320-f003:**
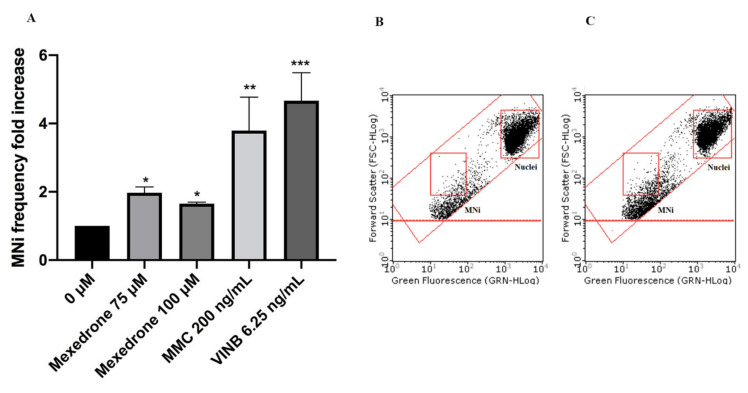
MNi frequency fold increase on TK6 cells after 26 h treatment with mexedrone at the indicated concentrations compared to the negative control [0 μM] and to positive controls [MMC and VINB] (**A**), plot of nuclei and MNi in the negative control (**B**) and in 100 µM mexedrone -treated cultures (**C**). Each bar represents the mean ± SEM of five independent experiments. Data were analyzed using repeated ANOVA followed by Bonferroni post-test. * *p* < 0.05 vs. 0 µM; ** *p* < 0.01 vs. 0 µM; *** *p* < 0.001 vs. 0 µM.

**Figure 4 ijms-22-06320-f004:**
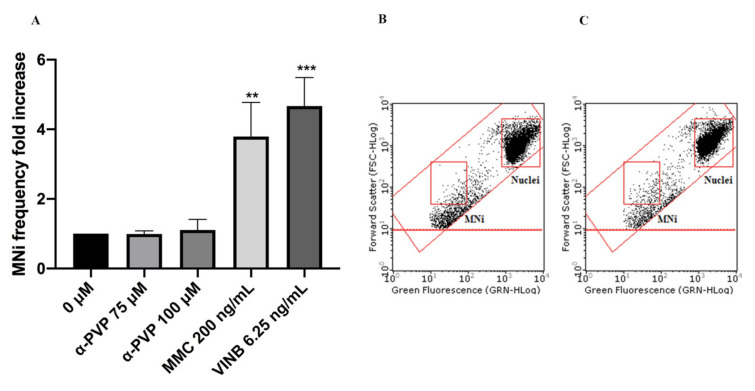
MNi frequency fold increase on TK6 cells after 26 h treatment with α-PVP at the indicated concentrations compared to the negative control [0 µM] and to positive controls [MMC and VINB] (**A**), plot of nuclei and MNi in the negative control (**B**) and in 100 µM α-PVP -treated cultures (**C**). Each bar represents the mean ± SEM of five independent experiments. Data were analyzed using repeated ANOVA followed by Bonferroni post-test. ** *p* < 0.01 vs. 0 µM; *** *p* < 0.001 vs. 0 µM.

**Figure 5 ijms-22-06320-f005:**
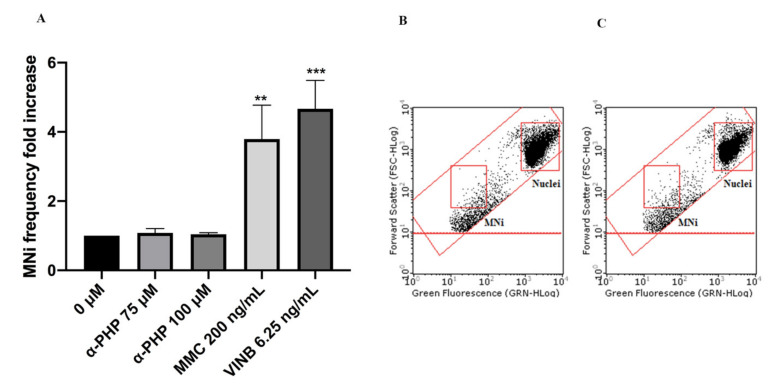
MNi frequency fold increase on TK6 cells after 26 h treatment with α-PHP at the indicated concentrations compared to the negative control [0 µM] and to positive controls [MMC and VINB] (**A**), plot of nuclei and MNi in the negative control (**B**) and in 100 µM α-PHP-treated cultures (**C**). Each bar represents the mean ± SEM of five independent experiments. Data were analyzed using repeated ANOVA followed by Bonferroni post-test. ** *p* < 0.01 vs. 0 µM; *** *p* < 0.001 vs. 0 µM.

**Figure 6 ijms-22-06320-f006:**
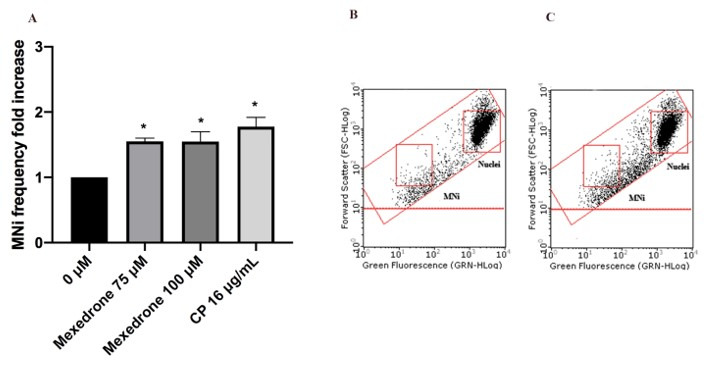
MNi frequency fold increase on TK6 cells after 3 h + S9 mix treatment with mexedrone at the indicated concentrations compared to the negative control [0 µM] and to the positive control [CP] (**A**), plot of nuclei and MNi in the negative control (**B**) and in 100 µM mexedrone-treated cultures (**C**). Each bar represents the mean ± SEM of five independent experiments. Data were analyzed using repeated ANOVA followed by Bonferroni post-test. * *p* < 0.05 vs. 0 µM.

**Figure 7 ijms-22-06320-f007:**
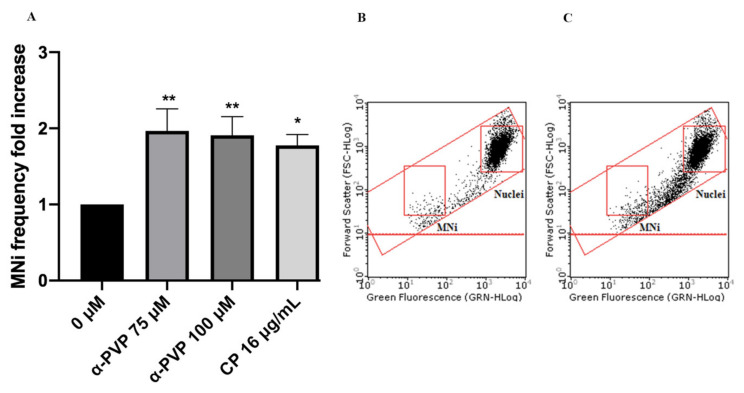
MNi frequency fold increase on TK6 cells after 3 h + S9 mix treatment with α-PVP at the indicated concentrations compared to the negative control [0 µM] and to the positive control [CP] (**A**), plot of nuclei and MNi in the negative control (**B**) and in 100 µM α-PVP-treated cultures (**C**). Each bar represents the mean ± SEM of five independent experiments. Data were analyzed using repeated ANOVA followed by Bonferroni post-test. * *p* < 0.05 vs. 0 µM; ** *p* < 0.01 vs. 0 µM.

**Figure 8 ijms-22-06320-f008:**
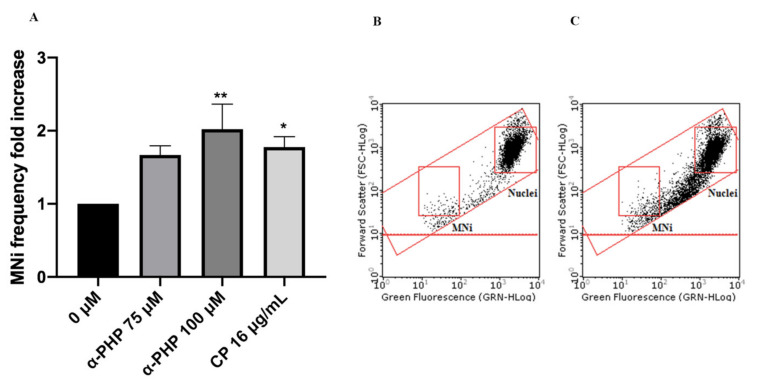
MNi frequency fold increase on TK6 cells after 3 h + S9 mix treatment with α-PHP at the indicated concentrations compared to the negative control [0 µM] and to the positive control [CP] (**A**), plot of nuclei and Mni in the negative control (**B**) and in 100 µM α-PHP-treated cultures (**C**). Each bar represents the mean ± SEM of five independent experiments. Data were analyzed using repeated ANOVA followed by Bonferroni post-test. * *p* < 0.05 vs. 0 µM; ** *p* < 0.01 vs. 0 µM.

**Figure 9 ijms-22-06320-f009:**
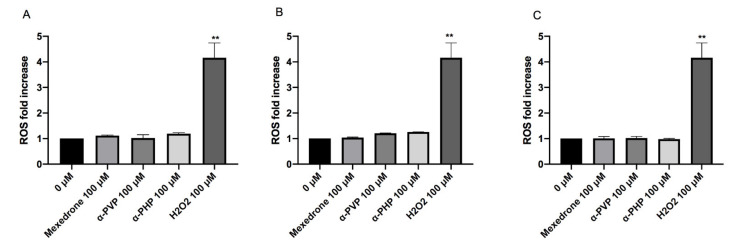
ROS fold increase on TK6 cells after 1 (**A**), 6 (**B**) and 12 (**C**) h treatment with mexedrone, α-PVP, α-PHP at the indicated concentrations compared to the negative control [0 µM] and to positive control (H_2_O_2_). Each bar represents the mean ± SEM of five independent experiments. Data were analyzed using repeated ANOVA followed by Bonferroni post-test. ** *p* < 0.01 vs. 0 µM.

**Table 1 ijms-22-06320-t001:** RPD on TK6 cells after 26 h treatment with mexedrone, α-PVP, α-PHP at the indicated concentrations compared to the negative control [0 μM]. Data are presented as mean ± SEM of five independent experiments.

Relative Population Doubling (RPD)			
	Mexedrone	α-PVP	α-PHP
**0 µM**	100.00%	100.00%	100.00%
**25 µM**	97.0% ± 3.2	91.5% ± 5.9	88.3% ± 1.3
**35 µM**	92.0% ± 4.2	88.4% ± 6.8	94.4% ± 1.7
**50 µM**	88.4% ± 7.9	88.7% ± 7.1	96.8% ± 3.2
**75 µM**	95.2% ± 3.5	89.9% ± 4.6	85.2% ± 3.5
**100 µM**	90.1% ± 3.6	87.6% ± 6.3	85.9% ± 5.6

## Data Availability

Data is available upon request.

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
