# Peer review of "Evaluation of Cytotoxic and Mutagenic Effects of the Synthetic Cathinones Mexedrone, α-PVP and α-PHP"

_ijms, 2021, doi:10.3390/ijms22126320_

Round 1

Reviewer 1 Report

The authors showed great knowledge of the subject. The research was carried out professionally.

Minor concerns:

1)The introduction and discussion are a little extense from my point of view but it is understandable that the authors wanted to provide the reader with as much information as possible on the topic.

2)There are some minor editing errors in the text, for example, line 98-single parenthesis.

Author Response

Reviewer: The authors showed great knowledge of the subject. The research was carried out professionally.

Author: We thank the Reviewer for his appreciation of our work and helpful suggestions to improve it. The changes made are highlighted in red in the manuscript for faster viewing.

Minor concerns:

Reviewer: 1) The introduction and discussion are a little extense from my point of view but it is understandable that the authors wanted to provide the reader with as much information as possible on the topic.

Author: We agree with the Reviewer, in fact we have tried to provide the reader with as much information as possible on the topic. We apologize if we keep this approach. Other references in this type of work have asked us to be exhaustive ...

Reviewer: 2) There are some minor editing errors in the text, for example, line 98-single parenthesis.

Author: We corrected all the typos.

Reviewer 2 Report

In nthe introduction Section, speaking about cathinones and their toxicity som papers should be referenced and briefly reported since they constitute the early studies :

Mephedrone related fatalities: a review.

Busardò FP, Kyriakou C, Napoletano S, Marinelli E, Zaami S.Eur Rev Med Pharmacol Sci. 2015 Oct;19(19):3777-90. Neurotoxicity Induced by Mephedrone: An up-to-date Review. Pantano F, Tittarelli R, Mannocchi G, Pacifici R, di Luca A, Busardò FP, Marinelli E.Curr Neuropharmacol. 2017;15(5):738-749 Synthetic cathinones related fatalities: an update. Zaami S, Giorgetti R, Pichini S, Pantano F, Marinelli E, Busardò FP.Eur Rev Med Pharmacol Sci. 2018 Jan;22(1):268-274 A Review of Synthetic Cathinone-Related Fatalities From 2017 to 2020. La Maida N, Di Trana A, Giorgetti R, Tagliabracci A, Busardò FP, Huestis MA.Ther Drug Monit. 2021 Feb 1;43(1):52-68.

-In consideration of the fact that NPS and therefore synthetic cathinones act mainly on the SNC, justify in a better way the use of Human TK6 lymphoblast cells.  

-Authors use a series of drug concentrations ranging between 25 and 100 µM. What is the relevance of these concentrations if compared to the level of synthetic cathinones detectable in the human serum upon intake?  

-Are there in vivo studies available reporting mutagenic effects of synthetic cathinones in mice or rats? If so, they should be discussed. 

-Authors investigated only the mutagenic activity of the metabolites produced by α-PVP and α-PHP but not of Mexedrone (as the parent compound was already mutagenic). 

However, it would be interesting to test also the mutagenic activity of Mexedrone using  S9 mix in order to evaluate whether an enhancement of the action or a detoxification / reduction is observed.

Author Response

We thank the Reviewer for helpful suggestions to improve our work. The required information and changes made, are listed point by point and highlighted in red in the manuscript for faster viewing.

Reviewer : 1) In the introduction Section, speaking about cathinones and their toxicity some papers should be referenced and briefly reported since they constitute the early studies:

-Mephedrone related fatalities: a review. Busardò FP, Kyriakou C, Napoletano S, Marinelli E, Zaami S.Eur Rev Med Pharmacol Sci. 2015 Oct;19(19):3777-90.

-Neurotoxicity Induced by Mephedrone: An up-to-date Review.Pantano F, Tittarelli R, Mannocchi G, Pacifici R, di Luca A, Busardò FP, Marinelli E.Curr Neuropharmacol. 2017;15(5):738-749

-Synthetic cathinones related fatalities: an update.Zaami S, Giorgetti R, Pichini S, Pantano F, Marinelli E, Busardò FP.Eur Rev Med Pharmacol Sci. 2018 Jan;22(1):268-274A

-Review of Synthetic Cathinone-Related Fatalities From 2017 to 2020.La Maida N, Di Trana A, Giorgetti R, Tagliabracci A, Busardò FP, Huestis MA.Ther Drug Monit. 2021 Feb 1;43(1):52-68.

Author: We included these references in the introduction section.

Reviewer: 2) In consideration of the fact that NPS and therefore synthetic cathinones act mainly on the SNC, justify in a better way the use of Human TK6 lymphoblast cells.

Author: As suggested by reviewer, we expanded the justification for choosing the TK6 cells in the discussion section (lines 237-239).

Reviewer: 3) Authors use a series of drug concentrations ranging between 25 and 100 µM. What is the relevance of these concentrations if compared to the level of synthetic cathinones detectable in the human serum upon intake?

Author: We discussed these points in discussion section (lines 336-342)

Reviewer: 4) Are there in vivo studies available reporting mutagenic effects of synthetic cathinones in mice or rats? If so, they should be discussed.

Author: As suggested by the reviewer, we inserted the comment required in the discussion section (lines 333-336)

Reviewer: 5) Authors investigated only the mutagenic activity of the metabolites produced by α-PVP and α-PHP but not of Mexedrone (as the parent compound was already mutagenic). However, it would be interesting to test also the mutagenic activity of Mexedrone using S9 mix in order to evaluate whether an enhancement of the action or a detoxification / reduction is observed.

Author: As rightly suggested by the reviewer we tested mexedrone also in the presence of S9 mix and entered the results obtained in the results section (Figure 6).

Reviewer 3 Report

In this study, the authors investigated the potential of three synthetic cathinones to induce mutagenic effects in in vitro. This study follows previous research with focus of mutagenicity of synthetic cannabinoids and psychedelic phenethylamines.

Comments:

In line 156, the authors write: “Overall, the obtained results allowed to select the concentrations to be used in the mutagenic analysis. In particular, 75 and 100 μM were tested for all synthetic cathinones”. However, 75 µM nor 100 µM did not differ from the other concentrations used. Therefore, it would be helpful if the authors could better explain why they decided to use these concentrations.

-The method section is too short and does not allow to reproduce the experiments. Even if previous publications are referred to, more information (amount of cells seeded, well plates, etc.) should be given.

-The manuscript should once again be corrected by either a language editing service or by a native English speaker. Overall, the manuscript is easy to follow but various sentences still either contain typos or do not make sense language-wise. Furthermore, “in vitro” is sometimes written in italics and sometimes not, words are wrongfully capitalized, the word “and” is missing in lists (line 147, legend of Figure 2), and so on.

-“SCs” is sometimes used as abbreviation for synthetic cathinones as well as for synthetic cannabinoids, which is somewhat confusing. The authors may therefore consider spelling out “synthetic cathinones” unless they really like to stick with the abbreviation. However, if the authors like to use this abbreviation, then it should at least be abbreviated consistenly (it is for instance spelled out again in line 157).

 -Words like “novel psychoactive substances”, “mexedrone”, “fetal bovine serum” should not be capitalized. This also applies for various other reagents like EDTE, PBS, and so on.

-Line 52: the authors write: “For this reason, we decided to address the evaluation of the mutagenic potential of the large group of stimulants, starting with four psychoactive phenethylamines, 2C-H, 2C-I, 2C-B e 25B-NBOMe” However, these substances are psychedelics rather than stimulants and the sentence should be corrected accordingly.

-Line 60: The authors write: “SCs, usually sold as “chemicals”, ‘bath salts’ or ‘plant food’,….” This was true in the beginning of appearance of  synthetic cathinones. However, are cathinones nowadays really still being sold as bath salts, plant food, etc.?

-Line 77: the word “probably” should be removed as theses cathinones are certainly used due their psychostimulant effects.

-Line 87&88: SERT IC50<10 µM is incorrect. It should be SERT IC50>10 µM.

-Line 336: “Phenethylamines” should be changed to “cathinones”.

Author Response

We thank the Reviewer for helpful suggestions to improve our work. The required information and changes made, are listed point by point and highlighted in red in the manuscript for faster viewing.

In this study, the authors investigated the potential of three synthetic cathinones to induce mutagenic effects in in vitro. This study follows previous research with focus of mutagenicity of synthetic cannabinoids and psychedelic phenethylamines.

Comments:

Reviewer: 1) In line 156, the authors write: “Overall, the obtained results allowed to select the concentrations to be used in the mutagenic analysis. In particular, 75 and 100 μM were tested for all synthetic cathinones”. However, 75 µM nor 100 µM did not differ from the other concentrations used. Therefore, it would be helpful if the authors could better explain why they decided to use these concentrations.

Author: Since we did not observe any cytotoxic and cytostatic effects and no significant induction of apoptosis at all the concentrations tested, we selected the highest one for the MN assay for all the substances.

Reviewer: 2) The method section is too short and does not allow to reproduce the experiments. Even if previous publications are referred to, more information (amount of cells seeded, well plates, etc.) should be given.

Author: As suggested by the reviewer we expanded the “Materials and Method” section.

Reviewer: 3) The manuscript should once again be corrected by either a language editing service or by a native English speaker. Overall, the manuscript is easy to follow but various sentences still either contain typos or do not make sense language-wise. Furthermore, “in vitro” is sometimes written in italics and sometimes not, words are wrongfully capitalized, the word “and” is missing in lists (line 147, legend of Figure 2), and so on.

Author: We corrected them and proofread the whole manuscript again.

Reviewer: 4) “SCs” is sometimes used as abbreviation for synthetic cathinones as well as for synthetic cannabinoids, which is somewhat confusing. The authors may therefore consider spelling out “synthetic cathinones” unless they really like to stick with the abbreviation. However, if the authors like to use this abbreviation, then it should at least be abbreviated consistenly (it is for instance spelled out again in line 157).

Author: We decided to spell out “synthetic cathinones” in the entire text to make it clearer.

Reviewer: 5) Words like “novel psychoactive substances”, “mexedrone”, “fetal bovine serum” should not be capitalized. This also applies for various other reagents like EDTE, PBS, and so on.

Author: We corrected them.

Reviewer: 6) Line 52: the authors write: “For this reason, we decided to address the evaluation of the mutagenic potential of the large group of stimulants, starting with four psychoactive phenethylamines, 2C-H, 2C-I, 2C-B e 25B-NBOMe” However, these substances are psychedelics rather than stimulants and the sentence should be corrected accordingly.

Author: We corrected the sentence as you suggested.

Reviewer: 7) Line 60: The authors write: “SCs, usually sold as “chemicals”, ‘bath salts’ or ‘plant food’,….” This was true in the beginning of appearance of synthetic cathinones. However, are cathinones nowadays really still being sold as bath salts, plant food, etc.?

Author: We agree with the Reviewer since nowadays are sold as research chemicals or in other forms. We removed the sentence “usually sold as “chemicals”, “bath salts” or “plant food”.

Reviewer: 8) Line 77: the word “probably” should be removed as theses cathinones are certainly used due their psychostimulant effects.

Author: We corrected it.

Reviewer: 9) Line 87&88: SERT IC50<10 µM is incorrect. It should be SERT IC50>10 µM.

Author: We apologize for the mistake and have corrected it.

Reviewer: 10) Line 336: “Phenethylamines” should be changed to “cathinones”.

Author: We corrected it.

Round 2

Reviewer 3 Report

The authors have adressed all my comments and in my opinion,  the manuscript is now suitable for publication.